# The Infinite Mixture of Infinite Gaussian Mixtures

**Halid Z. Yerebakan**
Department of
Computer and Information Science
IUPUI
Indianapolis, IN 46202
hzyereba@cs.iupui.edu

**Bartek Rajwa**
Bindley Bioscience Center
Purdue University
W. Lafayette, IN 47907
rajwa@cyto.purdue.edu

**Murat Dundar**
Department of
Computer and Information Science
IUPUI
Indianapolis, IN 46202
dundar@cs.iupui.edu

## Abstract

Dirichlet process mixture of Gaussians (DPMG) has been used in the literature for clustering and density estimation problems. However, many real-world data exhibit cluster distributions that cannot be captured by a single Gaussian. Modeling such data sets by DPMG creates several extraneous clusters even when clusters are relatively well-defined. Herein, we present the infinite mixture of infinite Gaussian mixtures ($I^2$GMM) for more flexible modeling of data sets with skewed and multi-modal cluster distributions. Instead of using a single Gaussian for each cluster as in the standard DPMG model, the generative model of $I^2$GMM uses a single DPMG for each cluster. The individual DPMGs are linked together through centering of their base distributions at the atoms of a higher level DP prior. Inference is performed by a collapsed Gibbs sampler that also enables partial parallelization. Experimental results on several artificial and real-world data sets suggest the proposed $I^2$GMM model can predict clusters more accurately than existing variational Bayes and Gibbs sampler versions of DPMG.

## 1 Introduction

The traditional approach to fitting a Gaussian mixture model onto the data involves using the well-known expectation-maximization algorithm to estimate component parameters [7]. The major limitation of this approach is the need to define the number of clusters in advance. Although there are several ways to predict the number of clusters in a data set in an offline manner, these techniques are in general suboptimal as they decouple the two interdependent tasks: predicting the number of clusters and predicting model parameters.

Dirichlet process mixture of Gaussians (DPMG), also known as the infinite Gaussian mixture model (IGMM), is a Gaussian mixture model (GMM) with a Dirichlet process (DP) prior defined over mixture components [8]. Unlike traditional mixture modeling, DPMG predicts the number of clusters while simultaneously performing model inference. In the DPMG model the number of clusters can arbitrarily grow to better accommodate data as needed. DPMG in general works well when the clusters are well-defined with Gaussian-like distributions. When the distributions of clusters are heavy-tailed, skewed, or multi-modal multiple mixture components per cluster may be needed for more accurate modeling of cluster data. Since there is no dependency structure in DPMG to asso-

ciate mixture components with clusters, additional mixture components produced during inference are all treated as independent clusters. This results in a suboptimal clustering of underlying data.

We propose the infinite mixture of IGMMs ($I^2$GMM) for more accurate clustering of data sets exhibiting skewed and multi-modal cluster distributions. The underlying generative model of $I^2$GMM employs a different DPMG for each cluster data. A dependency structure is imposed across individual DPMGs through centering of their base distibutions at one of the atoms of the higher level DP. This way individual cluster data are modeled by lower level DPs using one DPMG for each cluster and atoms defining the base distributions of individual clusters and cluster proportions are modeled by the higher level DP. Our model allows sharing of the covariance matrices across mixture components of the same DPMG. The data model, which is conjugate to the base distributions of both higher and lower level DPs, makes obtaining closed form solutions of posterior predictive distributions possible. We use a collapsed Gibbs sampler scheme for inference. Each scan of the Gibbs sampler involves two loops. One that iterates over individual data instances to sample component indicator variables and another one that iterates over components to sample cluster indicator variables. Conditioned on the cluster indicator variables, component indicator variables can be sampled in a parallel fashion, which significantly speeds up inference under certain circumstances.

## 2   Related Work

Dependent Dirichlet processes (DDP) have been studied in the literature for modeling collection of distributions that vary in time, in spatial region, in covariate space, or in grouped data settings (images, documents, biological samples). Previous work most related to the current work involves studies that investigate DDP in grouped data settings.

Teh et al. uses a hierarchical DP (HDP) prior over the base distributions of individual DP models to introduce a sharing mechanism that allows for sharing of atoms across multiple groups [15]. When each group data is modeled by a different DPMG this allows for sharing of the same mean vector and covariance matrix across multiple groups. Such a dependency may potentially be useful in a multi-group setting. However, when all data are contained in a single group as in the current study sharing the same mixture component across multiple cluster distributions leads to shared mixture components being statistically unidentifiable.

The HDP-RE model by Kim & Smyth [10] and transformed DP by Sudderth et al. [14] relaxes the exact sharing imposed by HDP to have a dependency structure between multiple groups that allow for components to share perturbed copies of atoms. Although such a sharing mechanism may be useful for modeling random variations in component parameters across multiple groups, it is not very useful for clustering data sets with skewed and multi-modal distributions. Both HDP-RE and transformed DP still model each group data by a single DPMG and suffer from the same drawbacks as DPMG when clustering data sets with skewed and multi-modal distributions.

The nested Dirichlet Pricess (nDP) by Rodriguez et al. [13] is a DP whose base distribution is in turn another DP. This model is introduced for modeling multi-group data sets where groups share not just individual mixture components as in HDP but the entire mixture model defined by a DPMG. nDP can be adapted to single group data sets with multiple clusters but with the restriction that each DPMG is shared only once to ensure identifiability. Such a restriction practically eliminates dependencies across DPMGs modeling different clusters and would not offer clustering property at the group level.

Unlike existing work which creates dependencies across multiple DPMG through exact or perturbed sharing of mixture components or through sharing of the entire mixture model, proposed $I^2$GMM model associates each cluster with a distinct atom of the higher level DP through centering of the base distribution of the corresponding DPMG at that atom. Thus, the higher level DP defines meta-clusters whereas lower level DPs model actual cluster data. Mixture components associated with the same DPMG have their own mean vectors but share the same covariance matrix. Apart from preserving the conjugacy of the data model covariance sharing across mixture components of the same DPMG allows for identification of clusters that differ in cluster shapes even when they are not well separated by their means.

# 3 Dirichlet Process Mixture

Dirichlet process is a distribution over discrete distributions. It is parameterized by a concentration parameter $\alpha$ and a base distribution $H$ denoted by $DP(\alpha H)$. Each probability mass in a sample discrete distribution is called as atom. According to the stick-breaking construction of DP [9], each sample from a DP can be considered as a collection of countably infinite number of atoms. In this representation base distribution is a prior over the locations of the atoms and concentration parameter affects the distribution of the atom weights, i.e., stick lengths. Another popular characterization of DP includes the Chinese restaurant process (CRP) [3] which we utilize during model inference. Discrete nature of its samples makes DP suitable as a prior distribution over mixture weights in mixture models. Although samples from DP are defined by an infinite dimensional discrete distribution, the posterior distribution conditioned on a finite data always uses finite number of mixture components.

We denote each data instance by $\boldsymbol{x}_i \in \mathbb{R}^d$ where $i \in \{1, ..., n\}$, $n$ is the total number of data instances. For each instance, $\theta_i$ indicates the set of parameters from which the instance is sampled. For the Gaussian data model $\theta_i = \{\boldsymbol{\mu}_i, \Sigma_i\}$ where $\boldsymbol{\mu}_i$ denotes the mean vector and $\Sigma_i$ the covariance matrix. The generative model of the Dirichlet Process Gaussian Mixture is given by (1).

$$
\begin{aligned}
\boldsymbol{x}_i &\sim p(\boldsymbol{x}_i|\theta_i) \\
\theta_i &\sim G \\
G &\sim DP(\alpha H)
\end{aligned}
\tag{1}
$$

Owing to the discreteness of the distribution $G$, $\theta_i$'s corresponding to different instances will not be all distinct. It is this property of DP that offers clustering over $\theta_i$ and in turn over data instances. Choosing $H$ from a family of distributions conjugate to the Gaussian distribution produces a closed-form solution for the posterior predictive distribution of DPMG. The bivariate prior over the atoms of $G$ is defined in (2).

$$
H = NIW(\boldsymbol{\mu_0}, \Sigma_0, \kappa_0, m) = N(\boldsymbol{\mu}|\boldsymbol{\mu_0}, \frac{\Sigma}{\kappa_0}) \times W^{-1}(\Sigma|\Sigma_0, m)
\tag{2}
$$

where $\boldsymbol{\mu}_0$ is the prior mean and $\kappa_0$ is a scaling constant that controls the deviation of the mean vectors from the prior mean. The parameter $\Sigma_0$ is the scaling matrix and $m$ is degrees of freedom. The posterior predictive distribution for a Gaussian data model and NIW prior can be obtained by integrating out $\boldsymbol{\mu}$ and $\Sigma$ analytically. Integrating out $\boldsymbol{\mu}$ and $\Sigma$ leaves us with the component indicator variables $t_i$ for each instance $x_i$ as the only random variables in the state space. Using the CRP representation of DP, $t_i$'s can be sampled as in (3).

$$
p(t_i = k|X, \boldsymbol{t}^{-i}) \quad \propto \quad \left\{
\begin{array}{ll}
\alpha p(\boldsymbol{x}_i) & \text{if } k = K + 1 \\
n_k^{-i} p(\boldsymbol{x}_i|A_k^{-i}, \bar{\boldsymbol{x}}_k^{-i}) & \text{if } k \leq K
\end{array}
\right\}
\tag{3}
$$

where $p(\boldsymbol{x}_i)$ and $p(\boldsymbol{x}_i|A_k, \bar{\boldsymbol{x}}_k)$ denote the posterior predictive distributions for an empty and occupied component, respectively, both of which are multivariate Student-t distributions. $X$ and $\boldsymbol{t}$ denote the sets of all data instances and their corresponding indicator variables, respectively. $n_k$ is the number of data instances in component $k$. $A_k$ and $\bar{\boldsymbol{x}}_k$ are the scatter matrix and sample mean for component $k$, respectively. The superscript $-i$ notation indicates the exclusion of the effect of instance $i$ from the corresponding variable. Inference for DPMG can also be performed using the stick-breaking representation of DP with the actual inference performed either by a Gibbs sampler or through variational Bayes [5, 11].

# 4 The Infinite Mixture of Infinite Gaussian Mixture Models

When modeling data sets containing skewed and multi-modal clusters, DPMG tends to produce multiple components for each cluster. Owing to the single-layer structure of DPMG, no direct associations among different components of the same cluster can be made. As a result of this limitation all components are treated as independent clusters resulting in a situation where the number of clusters are overpredicted and the actual cluster data are split into multiple subclusters. A more flexible model for clustering data sets with skewed and multi-modal clusters can be obtained using a two-

layer generative model as in (4).

$$
\begin{aligned}
\boldsymbol{x}_i &\sim N(\boldsymbol{x}_i|\boldsymbol{\mu}_i, \Sigma_j) \\
\boldsymbol{\mu}_i &\sim G_j \\
G_j &\sim DP(\alpha H_j) \\
H_j &= N(\boldsymbol{\mu}_j, \Sigma_j/\kappa_1) \\
(\boldsymbol{\mu}_j, \Sigma_j) &\sim G \\
G &\sim DP(\gamma H) \\
H &= NIW(\boldsymbol{\mu}_0, \Sigma_0, \kappa_0, m)
\end{aligned}
\tag{4}
$$

In this model, top layer DP generates cluster-specific parameters $\boldsymbol{\mu}_j$ and $\Sigma_j$ according to the base distribution $H$ and concentration parameter $\gamma$. These parameters in turn define the base distributions $H_j$ of the bottom layer DPs. Since each $H_j$ is representing a different cluster, $H_j$'s can be considered as meta-clusters from which mixture components of the corresponding cluster are generated. In this model both the number of clusters and the number of mixture components within a cluster can be potentially infinite hence the name $I^2$GMM. The top layer DP models the number of clusters, their sizes, and the base distribution of the bottom layer DPs whereas each bottom layer DP models the number of components in a cluster and their sizes. Allowing atom locations in the bottom layer DPGMs to be different than their corresponding cluster atom provides the flexibility to model clusters that cannot be effectively modeled by a single Gaussian. The scaling parameter $\kappa_1$ adjusts within cluster scattering of the component mean vectors whereas the scaling parameter $\kappa_0$ adjusts between cluster scattering of the cluster-specific mean vectors. Expressing both $H$ and $H_j$'s as functions of $\Sigma_j$ not only preserves the conjugacy of the model but also allows for sharing of the same covariance matrix across mixture components of the same cluster.

Posterior inference for the proposed model in (4) can be performed by a collapsed Gibbs sampler by iteratively sampling component indicator variables $\boldsymbol{t} = \{t_i\}_{i=1}^{n}$ of data instances and cluster indicator variables $\boldsymbol{c} = \{c_k\}_{k=1}^{K}$ of mixture components. When sampling $t_i$ we restrict sampling with components whose cluster indicator variables are equal to $c_{t_i}$ in addition to a new component. The conditional distribution for sampling $t_i$ can be expressed by the following equation.

$$
p(t_i = k|X, \boldsymbol{t}^{-i}, \boldsymbol{c}) \propto
\begin{cases}
\alpha p(\boldsymbol{x}_i) & if\ k = K+1 \\
n_k^{-i} p(\boldsymbol{x}_i|A_k^{-i}, \bar{\boldsymbol{x}}_k^{-i}, S_{c_k}) & if\ k : c_k = c_{t_i}
\end{cases}
\tag{5}
$$

where $S_{c_k} = \{A_\ell, \bar{\boldsymbol{x}}_\ell, n_\ell\}_{\ell:c_\ell=c_k}$. When sampling component indicator variables, owing to the dependency among data instances, removing a data instance from a component not only affect the parameters of the components it belongs to but also the corresponding cluster parameters. Technically speaking the parameters of both the component and corresponding cluster has to be updated for exact inference. However, updating cluster parameters for every data instance removed will significantly slow down inference. For practical purposes we only update component parameters and assume that removing a single data instance does not significantly change cluster parameters. The conditional distribution for sampling $c_k$ can be expressed by the following equation.

$$
p(c_k = j|X, \boldsymbol{t}, \boldsymbol{c}^{-k}) \propto
\begin{cases}
\gamma \prod_{i:t_i=k} p(\boldsymbol{x}_i) & if\ j = J+1 \\
m_j \prod_{i:t_i=k} p(\boldsymbol{x}_i|S_j) & if\ j \leq J
\end{cases}
\tag{6}
$$

where $S_j = \{A_\ell, \bar{\boldsymbol{x}}_\ell, n_\ell\}_{\ell:c_\ell=j}$, $J$ is the number of clusters, and $m_j$ is the number of mixture components assigned to cluster $j$. Next, we discuss the derivation of the component-level posterior predictive distributions, i.e., $p(\boldsymbol{x}_i|A_k^{-i}, \bar{\boldsymbol{x}}_k^{-i}, S_{c_k})$, which can be obtained by evaluating the integral in (7).

$$
p(\boldsymbol{x}_i|A_k^{-i}, \bar{\boldsymbol{x}}_k^{-i}, S_{c_k}) = \int \int p(\boldsymbol{x}_i|\boldsymbol{\mu}_k, \Sigma_{c_k}) p(\boldsymbol{\mu}_k, \Sigma_{c_k}|A_k^{-i}, \bar{\boldsymbol{x}}_k^{-i}, S_{c_k}) \partial \boldsymbol{\mu}_k \partial \Sigma_{c_k}
\tag{7}
$$

To evaluate the integral in (7) we need the posterior distribution of the component parameters, namely $p(\boldsymbol{\mu}_k, \Sigma_{c_k}|A_k^{-i}, \bar{\boldsymbol{x}}_k^{-i}, S_{c_k})$, which is proportional to

$$
\begin{aligned}
p(\boldsymbol{\mu}_k, \Sigma_{c_k}|A_k^{-i}, \bar{\boldsymbol{x}}_k^{-i}, S_{c_k}) &\propto p(\boldsymbol{\mu}_k, \Sigma_{c_k}, A_k^{-i}, \bar{\boldsymbol{x}}_k^{-i}|S_{c_k}) \\
&= p(\bar{\boldsymbol{x}}_k^{-i}|\boldsymbol{\mu}_k, \Sigma_{c_k}) p(A_k^{-i}|\Sigma_{c_k}) p(\boldsymbol{\mu}_k|\Sigma_{c_k}, S_{c_k}) p(\Sigma_k|S_{c_k})
\end{aligned}
\tag{8}
$$

where

$$
\begin{aligned}
p(\bar{\boldsymbol{x}}_k^{-i}|\boldsymbol{\mu}_k,\Sigma_{c_k}) &= N\left(\mu_k,(n_k^{-i})^{-1}\Sigma_{c_k}\right)\\
p(A_k^{-i}|\Sigma_{c_k}) &= W\left(\Sigma_{c_k},n_k^{-i}-1\right)\\
p(\boldsymbol{\mu}_k|\Sigma_{c_k},S_{c_k}) &= N(\bar{\boldsymbol{\mu}},\bar{\kappa}^{-1}\Sigma_{c_k})\\
p(\Sigma_{c_k}|S_{c_k}) &= W^{-1}\left(\Sigma_0+\sum_{\ell:c_\ell=c_k}A_\ell,m+\sum_{\ell:c_\ell=c_k}(n_\ell-1)\right)\\
\bar{\boldsymbol{\mu}} &= \frac{\sum_{\ell:c_\ell=c_k}\frac{n_\ell\kappa_1}{(n_\ell+\kappa_1)}\bar{\boldsymbol{x}}_\ell+\kappa_0\boldsymbol{\mu}_0}{\sum_{\ell:c_\ell=c_k}\frac{n_\ell\kappa_1}{(n_\ell+\kappa_1)}+\kappa_0}\\
\bar{\kappa} &= \frac{(\sum_{\ell:c_\ell=c_k}\frac{n_\ell\kappa_1}{(n_\ell+\kappa_1)}+\kappa_0)\kappa_1}{\sum_{\ell:c_\ell=c_k}\frac{n_\ell\kappa_1}{(n_\ell+\kappa_1)}+\kappa_0+\kappa_1}
\end{aligned}
$$

Once we substitute $p(\boldsymbol{\mu}_k,\Sigma_{c_k}|A_k^{-i},\bar{\boldsymbol{x}}_k^{-i},S_{c_k})$ into (7) and evaluate the integral we obtain $p(\boldsymbol{x}_i|A_k^{-i},\bar{\boldsymbol{x}}_k^{-i},S_{c_k})$ in the form of a multivariate Student-t distribution.

$$
p(\boldsymbol{x}_i|A_k^{-i},\bar{\boldsymbol{x}}_k^{-i},S_{c_k})=stu-t(\hat{\boldsymbol{\mu}},\hat{\Sigma},v)
\tag{9}
$$

The location vector ($\hat{\mu}$), the scale matrix ($\hat{\Sigma}$), and the degrees of freedom ($v$) are given below.

Location vector:

$$
\hat{\boldsymbol{\mu}}=\frac{n_k^{-i}\bar{\boldsymbol{x}}_k^{-i}+\bar{\kappa}\bar{\boldsymbol{\mu}}}{n_k^{-i}+\bar{\kappa}}
\tag{10}
$$

Scale matrix:

$$
\hat{\Sigma}=\frac{\Sigma_0+\sum_{\ell:c_\ell=c_k}A_\ell+A_k^{-i}+\frac{n_k^{-i}\bar{\kappa}}{n_k^{-i}+\bar{\kappa}}(\bar{\boldsymbol{x}}_k^{-i}-\bar{\boldsymbol{\mu}})(\bar{\boldsymbol{x}}_k^{-i}-\bar{\boldsymbol{\mu}})^T}{\frac{(\bar{\kappa}+n_k^{-i})\,v}{(\bar{\kappa}+n_k^{-i}+1)}}
\tag{11}
$$

Degrees of freedom:

$$
v=m+\sum_{\ell:c_\ell=c_k}(n_\ell-1)+n_k^{-i}-d+1
\tag{12}
$$

The cluster-level posterior predictive distributions can be readily obtained from $p(\boldsymbol{x}_i|A_k^{-i},\bar{\boldsymbol{x}}_k^{-i},S_{c_k})$ by dropping $A_k$, $\bar{\boldsymbol{x}}_k$, and $n_k$ from (10)-(12). Similarly, posterior predictive distribution for an empty component/cluster can be obtained by dropping $S_{c_k}$ from (10)-(12) in addition to $A_k$, $\bar{\boldsymbol{x}}_k$, and $n_k$.

Thanks to the two-layer structure of the proposed model, the inference for I²GMM can be partially parallelized. Conditioned on the cluster indicator variables, component indicator variables for data instances in the same cluster can be sampled independent of the data instances in other clusters. The amount of actual speed up that can be achieved by parallelization depends on multiple factors including the number of clusters, cluster sizes, and how fast the other loop that iterates over cluster indicator variables can be run.

## 5 Experiments

We evaluate the proposed I²GMM model on five different data sets and compare its performance against three different versions of DPMG in terms of clustering accuracy and run time.

### 5.1 Data Sets

*Flower formed by Gaussians:* We generated a flower-shaped two-dimensional artificial data set using a different Gaussian mixture model for each of the four different parts (petals, stem, and two leaves) of the flower. Each part is considered as a separate cluster. Although covariance matrices are same for all Gaussian components within a mixture they do differ between mixtures to create clusters of different shapes. Petals are formed by a mixture of nine Gaussians sharing a spherical covariance. Stem is formed by a mixture of four Gaussians sharing a diagonal covariance. Each leaf is formed by a mixture of two Gaussians sharing a full covariance. There are a total of seventeen Gaussian components, four clusters, and 17,000 instances (1000 instances per component) in this data set. Scatter plot of this data set is shown in Fig 1a.

*Lymphoma:* Lymphoma data set is one of the data sets used in the FlowCAP (Flow Cytometry Critical Assessment of Population Identification Methods) 2010 competition [1]. This data set consists

of thirty sub-data sets each generated from a lymph node biopsy sample of a patient using a flow cytometer. Flow cytometry is a single-cell screening, analysis, and sorting technology that plays a crucial role in research and clinical immunology, hematology, and oncology. The cellular phenotypes are defined in FC by combinations of morphological features (measured by elastic light scatter) and abundances of surface and intracellular markers revealed by fluorescently labeled antibodies. In the lymphoma data set each of the sub-data set contains thousands of instances with each instance representing a cell by a five-dimensional feature vector. For each sub-data set cell populations are manually gated by experts. Each sub-data has between two to four cell populations, i.e., clusters. Owing to the intrinsic mechanical and optical limitations of a flow cytometer, distributions of cell populations in the FC data end up being heavy-tailed or skewed, which makes their modeling by a single Gaussian highly impractical [12]. Although clusters in this data set are relatively well-defined accurate modeling of cell distributions is a challenge due to skewed nature of distributions.

*Rare cell populations:* This data set is a small subset of one of the data sets used in the FlowCAP 2012 competition [1]. The data set contains about 279,546 instances with each instance characterizing a white blood cell in a six-dimensional feature space. There are three clusters manually labeled by experts. This is an interesting data set for two reasons. First, clusters are highly unbalanced in terms of the number of instances belonging to each cluster. Two of the clusters, which are highly significant for measuring immunological response of the patient, are extremely rare. The ratios of the number of instances available from each of the two rare classes to the total number of instances are 0.0004 and 0.0005, respectively. Second, the third cluster, which contains all cells not belonging to one of the two rare-cell populations, has a distribution that is both skewed and multi-modal making it extremely challenging to recover its distribution as a single cluster.

*Hyperspectral imagery:* This data set is a flightline over a university campus. The hyperspectral data provides image data in 126 spectral bands in the visible and infrared regions. A total of 21,518 pixels from eight different land cover types are manually labeled. Some of the land cover types such as roof tops have multi-modal distributions. Cluster sizes are also relatively unbalanced with pixels belonging to roof tops constituting about one half of the labeled pixels. To reduce run time the dimensionality is reduced by projecting the original data onto its first thirty principal components. The data with reduced dimensionality is used in all experiments.

*Letter recognition:* This is a benchmark data set available through the UCI machine learning repository [4]. There are twenty six well-balanced clusters (one for each letter) in this data set.

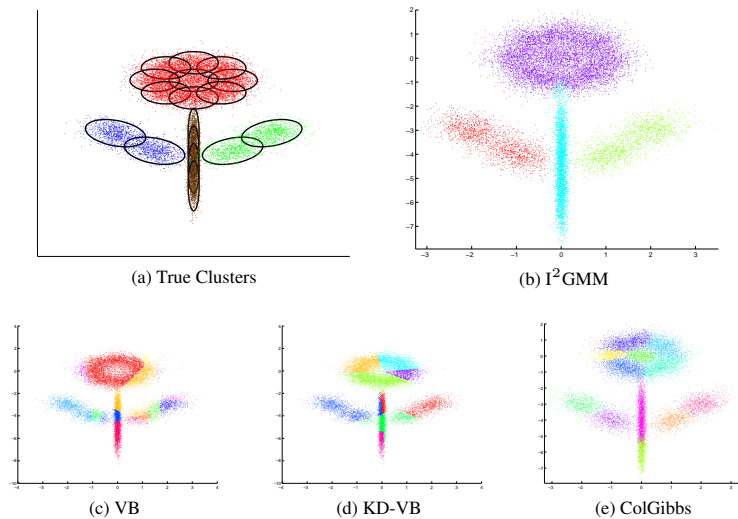

Figure 1: Clusters predicted by I²GMM, VB, KD-VB, and ColGibbs on the flower data set. Black contours in the first figure indicate distributions of individual Gaussian components forming the flower. Each color refers to a different cluster. Points denote data instances.

Table 1: Micro and macro $F_1$ scores produced by $I^2$GMM, VB, KD-VB, and ColGibbs on the five data sets. For each data set the first line includes micro $F_1$ scores and the second line macro $F_1$ scores. Numbers in parenthesis indicate standard deviations across ten repetitions. Results for the lyphoma data set are the average of results from thirty sub-data sets.

| Data set | $I^2$GMM | $I^2$GMMp | VB | KD-VB | ColGibbs |
|---|---|---|---|---|---|
| **Flower** | 0.975 (0.032) | 0.991 (0.003) | 0.640 (0.087) | 0.584 | 0.525 (0.010) |
| | 0.982 (0.015) | 0.990 (0.002) | 0.643 (0.059) | 0.639 | 0.611 (0.009) |
| **Lymphoma** | 0.920 (0.016) | 0.922 (0.020) | 0.454 (0.056) | 0.819 | 0.634 (0.034) |
| | 0.847 (0.021) | 0.847 (0.022) | 0.509 (0.044) | 0.762 | 0.656 (0.029) |
| **Rare classes** | 0.487 (0.031) | 0.493 (0.020) | 0.182 (0.015) | 0.353 | 0.234 (0.059) |
| | 0.756 (0.012) | 0.756 (0.010) | 0.441 (0.032) | 0.472 | 0.638 (0.023) |
| **Hyperspectral** | 0.624 (0.017) | 0.626 (0.021) | 0.433 (0.031) | 0.554 | 0.427 (0.024) |
| | 0.667 (0.018) | 0.661 (0.012) | 0.580 (0.034) | 0.380 | 0.596 (0.020) |
| **Letter Recognition** | 0.459 (0.015) | 0.467 (0.017) | 0.420 (0.015) | 0.267 | 0.398 (0.018) |
| | 0.460 (0.015) | 0.467 (0.017) | 0.420 (0.015) | 0.267 | 0.399 (0.018) |

## 5.2 Benchmark Models and Evaluation Metric

We compare the performance of the proposed $I^2$GMM model with three different versions of DPMG. These include the collapsed Gibbs sampler version (ColGibbs) discussed in Section 3, the variational Bayes version (VB) introduced in [5], and the KD-tree based accelerated variational Bayes version (KD-VB) introduced in [11]. For $I^2$GMM and ColGibbs we used our own implementations developed in C++. For VB and KD-VB we used existing MATLAB®(Natick, MA) implementations [1]. In order to see the effect of parallelization over execution times we ran the proposed technique in two modes: parallelized ($I^2$GMMp) and unparallelized ($I^2$GMM).

All data sets are scaled to have unit variance for each feature. The ColGibbs model has five free parameters $(\alpha, \Sigma_0, m, \kappa_0, \mu_0)$, $I^2$GMM model has two more parameters $(\kappa_1, \gamma)$ than ColGibbs. We use vague priors with $\alpha$ and $\gamma$ by fixing their value to one. We set $m$ to the minimum feasible value, which is $d+2$, to achieve maximum degrees of freedom in the shape of the covariance matrices. The prior mean $\mu_0$ is set to the mean of the entire data. The scale matrix $\Sigma_0$ is set to $I/s$, where $I$ is the identity matrix. This leaves the scaling constant $s$ of $\Sigma_0$, $\kappa_0$, and $\kappa_1$ as the three free parameters. We use $s = 150/(d(logd))$, $\kappa_0 = 0.05$, and $\kappa_1 = 0.5$ in experiments with all five data sets described above.

Micro and macro $F_1$ scores are used as performance measures for comparing clustering accuracy of these four techniques. As one-to-many matchings are expected between true and predicted clusters, the $F_1$ score for a true cluster is computed as the maximum of the $F_1$ scores for all predicted clusters. The Gibbs sampler for ColGibbs and $I^2$GMM are run for 1500 sweeps. The first 1000 samples are ignored as burn-in and eleven samples drawn with fifty sweeps apart are saved for final evaluation. We used an approach similar to the one proposed in [6] for matching cluster labels across different samples. The mode of cluster labels computed across ten samples are assigned as the final cluster label for each data instance. ColGibbs and $I^2$GMM use stochastic sampling whereas VB use a random initialization stage. Thus, these three techniques may produce results that vary from one run to other on the same data set. Therefore we repeat each experiment ten times and report average results of ten repetitions for these three techniques.

## 5.3 Results and Discussion

Micro and macro $F_1$ produced by the four techniques on all five data sets are reported in Table 1. On the flower data set $I^2$GMM achieves almost perfect micro and macro $F_1$ scores and correctly predicts the true number of clusters. The other three techniques produce several extraneous clusters which lead to poor $F_1$ scores. Clusters predicted by each of the four techniques are shown in Fig. 1. As expected ColGibbs identify distributions of individual Gaussian components as clusters as opposed to the actual clusters formed by mixtures of Gaussians. The piece-wise linear cluster boundaries

variational-dirichlet-process-gaussian-mixture-model

Table 2: Execution times for I²GMM, I²GMMp, VB, KD-VB, and ColGibbs in seconds on the five data sets. Numbers in parenthesis indicate standard deviations across ten repetitions. For the lymphoma data set results reported are average run-time per sub-data set.

| Data set | I²GMM | I²GMMp | VB | KD-VB | ColGibbs |
|---|---|---|---|---|---|
| **Flower** | 54 (2) | 41 (4) | 1 (0.2) | 7 | 59 (1) |
| **Lymphoma** | 119 (4) | 85 (4) | 51 (10) | 3 | 63 (3) |
| **Rare classes** | 9,738 (349) | 5,034 (220) | 2171 (569) | 16 | 7,250 (182) |
| **Hyperspectral** | 5,385 (109) | 3,456 (174) | 582 (156) | 2 | 7,455 (221) |
| **Letter Recognition** | 1545 (63) | 953 (26) | 122 (22) | 12 | 2,785 (123) |

obtained by VB and KD-VB, splitting original clusters into multiple subclusters, can be explained by simplistic model assumptions and approximations that characterize variational Bayes algorithms.

On the lymphoma data set the proposed I²GMM model achieves an average micro and macro $F_1$ scores of 0.920 and 0.848, respectively. These values are not only significantly higher than corresponding $F_1$ scores produced by the other three techniques but also on par with the best performing techniques in the FlowCAP 2010 competition [2]. Results for thirty individual sub-data sets in the lymphoma data set are available in the supplementary document. A similar trend is also observed with the other three real-world data sets as I²GMM achieves the best $F_1$ score among the four techniques. Between I²GMM and ColGibbs, I²GMM consistently generates less number of clusters across all data sets as expected. Overall, among the three different versions of DPMG that differ in the inference algorithm used, there is no clear consensus across five data sets as to which version predicts clusters more accurately. However, the proposed I²GMM model which extends DPMG to skewed and multi-modal clusters, clearly stands out as the most accurate model on all five data sets.

Run time results included in Table 2 favors variational Bayes techniques over the Gibbs sampler-based ones as expected. Despite longer run times, significantly higher $F_1$ scores achieved on data sets with diverse characteristics suggest that I²GMM can be preferred over DPMG for more accurate clustering. Results also suggest that I²GMM can benefit from parallelization. The actual amount of improvement in execution time depend on data characteristics as well as how fast the unparallelized loop can be run. The largest gain by parallelization is obtained on the rare classes data set which offered almost two-fold increase by parallelization on an eight-core workstation.

## 6    Conclusions

We introduced I²GMM for more effective clustering of multivariate data sets containing skewed and multi-modal clusters. The proposed model extends DPMG to introduce dependencies between components and clusters by a two-layer generative model. Unlike standard DPMG where each cluster is modeled by a single Gaussian, I²GMM offers the flexibility to model each cluster data by a mixture of potentially infinite number of components. Results on experiments with real and artificial data sets favor I²GMM over variational Bayes and collapsed Gibbs sampler versions of DPMG in terms of clustering accuracy. Although execution time can be improved by sampling component indicator variables in parallel, the amount of speed up that can be gained is limited with the execution time of the sampling of the cluster indicator variables. As most time consuming part of this task is the sequential computation of likelihoods for data instances, significant gains in execution time can be achieved by parallelizing the computation of likelihoods. I²GMM is implemented in C++. The source files and executables are available on the web. [2]

**Acknowledgments**

This research was sponsored by the National Science Foundation (NSF) under Grant Number IIS-1252648 (CAREER), by the National Institute of Biomedical Imaging and Bioengineering (NIBIB) under Grant Number 5R21EB015707, and by the PhRMA Foundation (2012 Research Starter Grant in Informatics). The content is solely the responsibility of the authors and does not represent the official views of NSF, NIBIB or PhRMA.

## Footnotes

[1]https://sites.google.com/site/kenichikurihara/academic-software/

[2]`https://github.com/halidziya/I2GMM`

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
