[Supplementary Material]

# The Infinite Mixture of Infinite Gaussian Mixtures
## *Supplementary Document*

**Halid Z. Yerebakan**
Department of
Computer and Information Science
IUPUI
Indianapolis, IN 46202
hzyereba@cs.iupui.edu

**Bartek Rajwa**
Bindley Bioscience Center
Purdue University
W. Lafayette, IN 47907
rajwa@cyto.purdue.edu

**Murat Dundar**
Department of
Computer and Information Science
IUPUI
Indianapolis, IN 46202
dundar@cs.iupui.edu

## 1 Data Set Characteristics

Table 1: Characteristics of the five data sets compared in terms of the number of clusters ($K$), the number of instances ($n$), the data dimensionality ($d$), the ratio of smallest to largest cluster size ($r$), skewedness, and multi-modality.

| Data set | K | n | d | r | skewed | multi-modal |
|---|---|---|---|---|---|---|
| **Flower** | 4 | 17,000 | 2 | 0.22 | no | yes |
| **Lymphoma** | 2-4 | 1,856-24,564 | 5 | 0.002-0.270 | yes | no |
| **Rare classes** | 3 | 279,546 | 6 | 0.0003 | yes | yes |
| **Hyperspectral** | 8 | 21,518 | 30 | 0.04 | no | yes |
| **Letter recognition** | 26 | 20,000 | 16 | 0.96 | unknown | unknown |

## 2 Number of Clusters Predicted

Table 2: Number of clusters generated by I$^2$GMM, I$^2$GMMp, VB, KD-VB, and ColGibbs on the five data sets. Predicted numbers shown are modes of ten repetitions. Clusters with less than ten instances are ignored. For the lymphoma data set the minimum and maximum of the predicted number of clusters across thirty sub-data sets are reported.

| Data set | True | I$^2$GMM | I$^2$GMMp | VB | KD-VB | ColGibbs |
|---|---|---|---|---|---|---|
| | | | Predicted | | | |
| **Flower** | 4 | 4 | 4 | 10 | 8 | 12 |
| **Lymphoma** | 2-4 | 2-9 | 2-9 | 17-53 | 2-8 | 5-22 |
| **Rare Classes** | 3 | 29 | 30 | 60 | 11 | 65 |
| **Hyperspectral** | 8 | 42 | 41 | 9 | 4 | 91 |
| **Letter Recognition** | 26 | 46 | 47 | 93 | 18 | 153 |

# 3 Results for the Thirty Sub-data Sets in the Lymphoma Data Set

Table 3: Micro $F_1$ scores produced by the four techniques on each of the thirty sub-data of the lymphoma data set. Numbers in parenthesis indicate standard deviations across ten repetitions.

| Sub-data set | $I^2$GMM | $I^2$GMMp | VB | KD-VB | Gibbs |
|---|---|---|---|---|---|
| 1 | 0.853 (0.005) | 0.853 (0.004) | 0.502 (0.043) | 0.690 | 0.705 (0.046) |
| 2 | 0.994 (0.001) | 0.994 (0.001) | 0.341 (0.048) | 0.980 | 0.551 (0.062) |
| 3 | 0.971 (0.037) | 0.969 (0.029) | 0.501 (0.087) | 0.926 | 0.823 (0.016) |
| 4 | 0.834 (0.024) | 0.834 (0.025) | 0.490 (0.047) | 0.802 | 0.588 (0.046) |
| 5 | 0.956 (0.001) | 0.955 (0.002) | 0.547 (0.049) | 0.947 | 0.827 (0.008) |
| 6 | 0.984 (0.002) | 0.984 (0.002) | 0.402 (0.044) | 0.985 | 0.626 (0.089) |
| 7 | 0.949 (0.008) | 0.951 (0.008) | 0.553 (0.062) | 0.722 | 0.627 (0.013) |
| 8 | 0.909 (0.009) | 0.914 (0.007) | 0.436 (0.042) | 0.670 | 0.611 (0.046) |
| 9 | 0.946 (0.003) | 0.947 (0.002) | 0.622 (0.097) | 0.934 | 0.847 (0.025) |
| 10 | 0.968 (0.002) | 0.967 (0.003) | 0.540 (0.058) | 0.941 | 0.662 (0.028) |
| 11 | 0.792 (0.102) | 0.856 (0.070) | 0.597 (0.056) | 0.864 | 0.874 (0.015) |
| 12 | 0.868 (0.022) | 0.867 (0.026) | 0.542 (0.059) | 0.803 | 0.709 (0.042) |
| 13 | 0.859 (0.002) | 0.859 (0.001) | 0.441 (0.045) | 0.752 | 0.578 (0.039) |
| 14 | 0.855 (0.023) | 0.854 (0.021) | 0.360 (0.051) | 0.770 | 0.573 (0.059) |
| 15 | 0.888 (0.015) | 0.890 (0.011) | 0.277 (0.031) | 0.690 | 0.435 (0.017) |
| 16 | 0.953 (0.013) | 0.955 (0.017) | 0.421 (0.076) | 0.920 | 0.740 (0.050) |
| 17 | 0.876 (0.012) | 0.862 (0.009) | 0.424 (0.053) | 0.743 | 0.607 (0.032) |
| 18 | 0.922 (0.013) | 0.916 (0.019) | 0.391 (0.056) | 0.747 | 0.517 (0.024) |
| 19 | 0.990 (0.001) | 0.990 (0.001) | 0.491 (0.091) | 0.975 | 0.682 (0.058) |
| 20 | 0.967 (0.014) | 0.964 (0.015) | 0.443 (0.051) | 0.785 | 0.592 (0.047) |
| 21 | 0.945 (0.010) | 0.945 (0.008) | 0.429 (0.046) | 0.806 | 0.627 (0.019) |
| 22 | 0.956 (0.009) | 0.958 (0.008) | 0.423 (0.093) | 0.713 | 0.616 (0.039) |
| 23 | 0.962 (0.002) | 0.962 (0.002) | 0.456 (0.043) | 0.744 | 0.632 (0.019) |
| 24 | 0.948 (0.007) | 0.951 (0.010) | 0.551 (0.055) | 0.892 | 0.666 (0.017) |
| 25 | 0.916 (0.025) | 0.907 (0.030) | 0.421 (0.048) | 0.730 | 0.559 (0.010) |
| 26 | 0.936 (0.010) | 0.932 (0.018) | 0.494 (0.054) | 0.806 | 0.641 (0.044) |
| 27 | 0.871 (0.049) | 0.866 (0.064) | 0.365 (0.024) | 0.828 | 0.480 (0.024) |
| 28 | 0.977 (0.009) | 0.971 (0.015) | 0.295 (0.059) | 0.907 | 0.440 (0.040) |
| 29 | 0.944 (0.004) | 0.946 (0.004) | 0.465 (0.061) | 0.723 | 0.632 (0.030) |
| 30 | 0.814 (0.034) | 0.838 (0.033) | 0.392 (0.039) | 0.782 | 0.551 (0.026) |
| Avg. | 0.920 (0.016) | 0.922 (0.020) | 0.454 (0.056) | 0.819 | 0.634 (0.034) |

Table 4: Macro $F_1$ scores produced by the four techniques on each of the thirty sub-data of the lymphoma data set. Numbers in parenthesis indicate standard deviations across ten repetitions.

| Sub-data set | $I^2$GMM | $I^2$GMMp | VB | KD-VB | ColGibbs |
|---|---|---|---|---|---|
| 1 | 0.840 (0.005) | 0.842 (0.004) | 0.504 (0.041) | 0.707 | 0.712 (0.039) |
| 2 | 0.967 (0.005) | 0.969 (0.004) | 0.552 (0.032) | 0.856 | 0.697 (0.040) |
| 3 | 0.963 (0.066) | 0.977 (0.016) | 0.635 (0.085) | 0.932 | 0.856 (0.016) |
| 4 | 0.670 (0.024) | 0.671 (0.024) | 0.461 (0.034) | 0.640 | 0.551 (0.030) |
| 5 | 0.702 (0.003) | 0.705 (0.013) | 0.559 (0.042) | 0.792 | 0.765 (0.037) |
| 6 | 0.971 (0.002) | 0.972 (0.002) | 0.465 (0.030) | 0.957 | 0.716 (0.055) |
| 7 | 0.855 (0.024) | 0.856 (0.020) | 0.563 (0.059) | 0.602 | 0.592 (0.012) |
| 8 | 0.897 (0.007) | 0.903 (0.009) | 0.444 (0.043) | 0.687 | 0.628 (0.036) |
| 9 | 0.938 (0.004) | 0.939 (0.002) | 0.607 (0.092) | 0.926 | 0.848 (0.034) |
| 10 | 0.978 (0.003) | 0.976 (0.005) | 0.616 (0.047) | 0.911 | 0.719 (0.017) |
| 11 | 0.618 (0.080) | 0.671 (0.055) | 0.529 (0.048) | 0.677 | 0.756 (0.021) |
| 12 | 0.797 (0.035) | 0.795 (0.042) | 0.505 (0.039) | 0.747 | 0.684 (0.029) |
| 13 | 0.677 (0.011) | 0.678 (0.008) | 0.561 (0.046) | 0.621 | 0.610 (0.066) |
| 14 | 0.831 (0.021) | 0.834 (0.022) | 0.532 (0.032) | 0.716 | 0.649 (0.030) |
| 15 | 0.784 (0.021) | 0.777 (0.025) | 0.345 (0.035) | 0.655 | 0.489 (0.010) |
| 16 | 0.898 (0.033) | 0.905 (0.049) | 0.581 (0.033) | 0.809 | 0.759 (0.020) |
| 17 | 0.787 (0.019) | 0.768 (0.015) | 0.384 (0.036) | 0.635 | 0.532 (0.026) |
| 18 | 0.744 (0.042) | 0.738 (0.013) | 0.482 (0.035) | 0.775 | 0.643 (0.014) |
| 19 | 0.892 (0.007) | 0.893 (0.009) | 0.536 (0.043) | 0.784 | 0.707 (0.029) |
| 20 | 0.933 (0.015) | 0.933 (0.014) | 0.431 (0.037) | 0.730 | 0.553 (0.042) |
| 21 | 0.896 (0.023) | 0.895 (0.018) | 0.479 (0.044) | 0.815 | 0.596 (0.032) |
| 22 | 0.887 (0.031) | 0.914 (0.026) | 0.518 (0.049) | 0.814 | 0.704 (0.029) |
| 23 | 0.916 (0.002) | 0.916 (0.003) | 0.515 (0.043) | 0.704 | 0.658 (0.023) |
| 24 | 0.921 (0.010) | 0.927 (0.015) | 0.508 (0.043) | 0.811 | 0.634 (0.022) |
| 25 | 0.862 (0.037) | 0.848 (0.039) | 0.417 (0.044) | 0.751 | 0.531 (0.012) |
| 26 | 0.909 (0.015) | 0.904 (0.026) | 0.524 (0.064) | 0.798 | 0.665 (0.048) |
| 27 | 0.850 (0.025) | 0.850 (0.037) | 0.400 (0.035) | 0.771 | 0.507 (0.020) |
| 28 | 0.881 (0.004) | 0.834 (0.092) | 0.572 (0.039) | 0.953 | 0.656 (0.019) |
| 29 | 0.794 (0.011) | 0.801 (0.009) | 0.513 (0.054) | 0.595 | 0.608 (0.021) |
| 30 | 0.740 (0.056) | 0.728 (0.043) | 0.523 (0.027) | 0.700 | 0.643 (0.044) |
| Avg. | 0.847 (0.021) | 0.847 (0.022) | 0.509 (0.044) | 0.762 | 0.656 (0.029) |

Table 5: Number of clusters generated by the four techniques on each of the thirty sub-data of the lymphoma data set. Predicted numbers shown are modes of ten repetitions.

| Sub-data set | $I^2$GMM | $I^2$GMMp | VB | KD-VB | ColGibbs |
|---|---|---|---|---|---|
| | True | | Predicted | | |
| 1 | 2 | 3 | 3 | 22 | 4 | 6 |
| 2 | 2 | 3 | 3 | 29 | 3 | 5 |
| 3 | 2 | 2 | 3 | 20 | 3 | 7 |
| 4 | 3 | 2 | 2 | 21 | 3 | 8 |
| 5 | 3 | 2 | 2 | 24 | 4 | 6 |
| 6 | 2 | 3 | 3 | 28 | 3 | 7 |
| 7 | 3 | 3 | 3 | 17 | 5 | 7 |
| 8 | 2 | 3 | 3 | 27 | 5 | 8 |
| 9 | 2 | 3 | 3 | 22 | 3 | 5 |
| 10 | 2 | 3 | 3 | 21 | 2 | 8 |
| 11 | 3 | 2 | 3 | 23 | 3 | 5 |
| 12 | 4 | 5 | 4 | 35 | 7 | 13 |
| 13 | 4 | 4 | 4 | 35 | 6 | 13 |
| 14 | 3 | 6 | 7 | 53 | 7 | 22 |
| 15 | 3 | 6 | 7 | 46 | 5 | 15 |
| 16 | 3 | 5 | 4 | 31 | 6 | 10 |
| 17 | 4 | 5 | 5 | 44 | 5 | 19 |
| 18 | 3 | 4 | 4 | 40 | 6 | 10 |
| 19 | 3 | 3 | 3 | 28 | 3 | 9 |
| 20 | 3 | 4 | 5 | 37 | 7 | 14 |
| 21 | 3 | 4 | 4 | 40 | 6 | 12 |
| 22 | 3 | 3 | 4 | 38 | 5 | 10 |
| 23 | 4 | 5 | 5 | 36 | 8 | 13 |
| 24 | 4 | 8 | 7 | 48 | 5 | 21 |
| 25 | 3 | 5 | 6 | 48 | 8 | 19 |
| 26 | 3 | 5 | 5 | 42 | 8 | 15 |
| 27 | 3 | 9 | 8 | 44 | 6 | 19 |
| 28 | 2 | 3 | 2 | 41 | 5 | 9 |
| 29 | 3 | 5 | 5 | 25 | 7 | 12 |
| 30 | 4 | 6 | 4 | 33 | 6 | 13 |