[Reviews · NeurIPS 2014]

Submitted by Assigned_Reviewer_2

The authors present a novel non-parametric Bayesian model for unsupervised clustering. The model uses a two level hierarchy of Dirichlet process priors to handle clusters which may be multi-modal, skewed and/or heavy tailed. The authors present a collapsed Gibbs sampler for inference which exploits the conjugacy of the model.

The authors do an excellent job of motivating the model by explaining the deficiencies of the standard infinite mixture of Gaussians. The review of previous works is good and the model is clearly presented and explained. The model is novel, though conceptually quite similar to the hierarchical Dirichlet process.

The authors have chosen an excellent selection of synthetic and real world datasets to benchmark their method. The performance of the method seems to be quite good, in particular the FlowCAP results seem competitive with methods which were tailored to the problem domain.

The authors only compare the model against the infinite Gaussians mixture model, with inference done using Gibbs sampling or VB. The paper could be improved if the authors compared against a richer set of models. In particular the authors may want to consider infinite mixtures of student-t's and skew normals to better clarify the advantages their model provides versus distributions which are more flexible than a single Gaussian emission. In addition other probabilistic models for clustering such as the one presented by Iwata et al. [1] are closer to the state of the art for clustering datasets with complex emission distributions.

Another minor concern is that authors fix the concentration parameter for the Dirichlet process priors (line 308). I would expect that the model would be somewhat sensitive to the values of these parameters, particularly the relative values at the two levels. It would be helpful if the authors could explore performance on the synthetic dataset across a range of values to determine robustness to these parameters. In addition the method in [2] could readily be applied to place priors on these values instead of fixing them.

Another very minor improvement would be for the authors to add details of the distributions and parameterisations they are using in the supplemental material. It would also be good to include explicit equations for the parameter updates described in lines 234-236.

[1] Tomoharu Iwata, David Duvenaud, and Zoubin Ghahramani. Warped mixtures for nonparametric cluster shapes. In 29th Conference on Uncertainty in Artificial Intelligence, Bellevue, Washington, July 2013.

[2] West, Mike. Hyperparameter estimation in Dirichlet process mixture models. Duke University, 1992.
Summary: A well written paper that presents a novel method with a well selected set of benchmark datasets. The model shows good performance on both synthetic and real world datasets.

Submitted by Assigned_Reviewer_7

One of the major problems in using Gaussian Mixture models for clustering is that in real world applications clusters often exhibit complex shapes which are poorly modeled by a single Gaussian. To overcome this problem the authors propose a hierarchical DP mixtures where single clusters are modeled via a DP mixtures.

The paper is clear and original and the performance of the model is assessed on several synthetic and real data examples. On the other hand, the authors compare their model with only DPMG models. It would be nice to see how the algorithm compares to other clustering algorithms.

Minor comment
In (4) notation is misleading. Better to use two different letters for mu_i and mu_j.
Summary: Nice paper which suggests a novel algorithm to better identify clusters in DP mixtures. It would be nice to see comparison with other clustering algorithms besides DPMG.

Submitted by Assigned_Reviewer_28

This paper present 'The Infinite Mixture of Infinite Gaussian Mixtures', a Bayesian nonparametric model consisting of two levels of Dirichlet process mixture model. This is introduced in the context of clustering, so that he distibution for a single cluster is represented by a DP mixture of Gaussians, thus allowing for more flexible cluster 'shapes'. Might also be of interest for finding sub-structure in clustering, cf hierarchical clustering methods.

The method is well-motivated and well-presented. The authors produce comparisons to several other comparable methods, running numerical experiments on a range of data sets. The results show that the proposed method is certainly doing soemthing new and interesting, extracting structure from the data that would otherwise be missed.

Encouragingly, the added complexity of the proposed model doesn't seem to onorous (in terms of the maths), and the run times reflect this. Being able to add richness to a class of models without making the complexity unwieldy is admirable.

Overall, I think this is a useful and interesting addition to the class of Dirichlet process related models, and one which has clear utility for a range of clustering tasks.

Minor points:

Is is worth also considering adjusted Rand index as a metric? This might give additional insights into the performance of the algorithms.

Table 1 - highlight best score for each data set in bold?

I think a clearer definition of micro- and macro- F1 scores would be useful to the read.
Summary: Overall, I think this is a useful and interesting addition to the class of Dirichlet process related models, and one which has clear utility for a range of clustering tasks.
Author Feedback
Author rebuttal: We thank all three reviewers for the wonderful comments and critiques. Our responses to critiques are as follows.

Reviewer 1:

1. Use infinite mixtures of student-t's and skewed normals to better clarify the advantages.
We have previously tried finite mixture of skewed-t distributions with the parameters estimated by EM and number of clusters selected by the scale-free weighted criterion but this approach underestimated the number of true clusters. The infinite version might generate better results than its finite counterpart but regrettably the one-week period assigned for rebuttals was not long enough to evaluate and test such an algorithm. We hope to include results from the infinite mixture of skewed-t distributions in a future version of this manuscript.

2. Fixing of the concentration parameters for Dirichlet process priors
We have preferred fixed values for concentration parameters (as opposed to the sampling approach used in Mike West's article) as we didn't want to further increase the number of design parameters by the hyperparameters of the Gamma priors required for sampling the concentration parameters. We do agree that sampling the concentration parameters may offer robustness but using hyperpriors for alpha and beta would introduce four additional hyperparameters on top of already existing ones.

3. Include explicit equations for the parameter updates described in lines 234-236

We will make sure that the supplementary material includes detailed derivations of these equations in the final version of the manuscript (if accepted).

Reviewer 2:

1. Adjusted rand index (ARI) might give additional insight into the performance of the algorithms.

We agree that ARI might offer additional insights about algorithms. We will include a new table containing ARI results in the final version of the manuscript (if accepted).

Reviewer 3:

1. It would be nice to see comparison with other clustering algorithms besides DPMG.

Since our algorithm was proposed as an improvement over the infinite Gaussian mixture model (IGMM) we limited our experiments with versions of IGMM implemented with different inference algorithms. We hope to evaluate the proposed algorithm in the context of a more broader class of clustering algorithms in a future version of this manuscript.